# Donated Blood Screening for HIV, HCV and HBV by ID-NAT and the Residual Risk of Iatrogenic Transmission in a Tertiary Care Hospital Blood Bank in Puebla, Mexico

**DOI:** 10.3390/v15061331

**Published:** 2023-06-06

**Authors:** Francisca Sosa-Jurado, Roxana Palencia-Lara, Cinthia Xicoténcatl-Grijalva, Maribel Bernal-Soto, Álvaro Montiel-Jarquin, Yolanda Ibarra-Pichardo, Nora Hilda Rosas-Murrieta, Rosalia Lira, Paulina Cortes-Hernandez, Gerardo Santos-López

**Affiliations:** 1Laboratorio de Biología Molecular y Virología, Centro de Investigación Biomédica de Oriente, Instituto Mexicano del Seguro Social, Metepec, Atlixco, Puebla 74360, CP, Mexico; sosajurado@hotmail.com (F.S.-J.); cingrijalva@gmail.com (C.X.-G.); 2Banco de Sangre, Hospital Especialidades, Unidad Médica de Alta Especialidad, Centro Médico Nacional General de División Manuel Ávila Camacho, Instituto Mexicano del Seguro Social, Puebla, Puebla 72000, CP, Mexico; roxana.palencia@imss.gob.mx (R.P.-L.); wikacrew_18@hotmail.com (M.B.-S.); maria.ibarrap@imss.gob.mx (Y.I.-P.); 3Coordinación Clínica de Investigación y Enseñanza en Salud, Hospital Especialidades, Unidad Médica de Alta Especialidad, Centro Médico Nacional General de División Manuel Ávila Camacho, Instituto Mexicano del Seguro Social, Puebla, Puebla 72000, CP, Mexico; alvaro.montielj@imss.gob.mx; 4Centro de Química, Instituto de Ciencias, Benemérita Universidad Autónoma de Puebla, Puebla, Puebla 72570, CP, Mexico; 5Unidad de Investigación Médica en Enfermedades Infecciosas y Parasitarias, UMAE Hospital de Pediatría, CMN Siglo XXI, Instituto Mexicano del Seguro Social, Mexico City 06720, MX, Mexico; rolica36@yahoo.com; 6Laboratorio de Metadinámica y Salud de Poblaciones, Centro de Investigación Biomédica de Oriente, Instituto Mexicano del Seguro Social (IMSS), Metepec 74360, MX, Mexico; paulina.cortes.hernandez@gmail.com

**Keywords:** blood safety, residual risk, hepatitis B virus, hepatitis C virus, human immunodeficiency virus

## Abstract

Hepatitis C virus (HCV), human immunodeficiency virus (HIV) and hepatitis B virus (HBV) can be transmitted by blood transfusion. Most transmission occurs during the acute viremic phase (AVP), before antibody development. To reduce transmission risk, individual donor nucleic acid testing (ID-NAT) is used. In Puebla, Mexico, serological tests and ID-NAT have been applied to screen blood donors and detect individuals in AVP. In the present study, 106,125 blood donors’ data in two periods (2012–2015 and 2017–2019) were analyzed. The residual risk (RR) values were calculated considering ID-NAT results. The RR for HIV was 14 in 1 million donations or 1 in 71,428, the RR for HVC was 6.8 in 1 million donations or 1 in 147,058 and, for HBV, it was 156 in 1 million donations, or 1 in 6410. Previously, it was predicted that the transmission RR of these viruses would be reduced in Mexico through better screening with NAT. The use of ID-NAT has, indeed, increased the safety of blood reserves for HIV and HCV. However, more research is needed to determine why the residual risk of HBV did not decrease as much over the study period. ID-NAT is an important complementary tool for blood donor screening that should be implemented.

## 1. Introduction

Hepatitis C virus (HCV), human immunodeficiency virus (HIV) and hepatitis B virus (HBV) can be transmitted by blood transfusion. The tests used to identify these viruses in donor blood are typically serology screenings, whose sensitivity and specificity have improved over time to prevent iatrogenic transmission [1]. Most of this transmission comes from individuals in the early acute viremic phase (AVP) that have not yet developed antibodies towards the infection and are therefore not reactive to serology screens. To detect them, additional screening methods by viral nucleic acid amplification tests (NAT) for HIV and HCV were first implemented in German blood banks in the 1990s [2], followed by other countries [3], with the main goal of reducing the window period, by the direct detection of viral RNA or DNA in individuals not yet reactive in serology [4]. Thus, NAT in blood banks has been considered complementary to serology screening [5]. NAT was initially performed in serum pools but has evolved to individual donor testing (ID-NAT). NAT can also aid to prevent: HBV transmission from persistent occult HBV infection (OBI) where HBsAg is not detectable serologically, but viral DNA is present in very low quantities [6]; or HIV transmission from donors with effective HIV antiretroviral therapy [7]. Despite serology and NAT screens to improve blood safety, a residual risk of iatrogenic pathogen transmission persists and needs to be analyzed.

In the last 10 years, public Mexican blood banks have progressively implemented NAT [8,9,10], and the first scientific report of its use in Mexico was in 2009, with the detection of five blood donors (BD) in the window period, 1 with HIV, 2 with HCV and 2 with HBV, in 1962 blood bank samples [11].

Only around 2.5–5% of BD in Mexico are voluntary or altruistic. The rest are replacement donations, requested from hospitalized patients who need transfusions and are frequently relatives or friends of the hospitalized [12]. The infections detected in replacement BD could be representative of the general population.

The aim of this study was to detect BD in the acute viremia phase (AVP), using the agreement between serological screening and ID-NAT. We also calculated the different residual risks of infection of these viruses through transfusion after the implementation of ID-NAT tests.

## 2. Materials and Methods

### 2.1. Blood Donors

Data were collected from serological tests for HIV, HCV and HBV of individuals who met the inclusion criteria to be BDs, by the Mexican regulation NM-253-SSA1-2012 [13], in the periods 2012–2015 and 2017–2019, from the blood bank at Mexican Social Security Institute (IMSS)—National Health Centre “Manuel Avila Camacho” (CMN-MAC). This is a large blood bank that manages over 20 thousand donors per year and is located in the main IMSS hospital for public tertiary level healthcare in Puebla city, the fourth largest metropolitan area in Mexico, with over 3.2 million inhabitants. The main criteria considered for donor eligibility in accordance with Mexican regulations were as follows: individuals must be between 18 and 65 years old, weigh at least 50 kg, not be experiencing illness on the day of donation, not have undergone surgery within the past six months, not having had tattoos, piercings, or acupuncture in the last twelve months, no history of or risk of hepatitis B or C, HIV-AIDS, Chagas disease, malaria, syphilis, or transmissible spongiform encephalopathies, such as Creutzfeld-Jakob syndrome, not having epilepsy, tuberculosis, or severe heart disease, not having received organ transplants, and not having used intravenous or inhaled drugs. For female donors, additional criteria include not being pregnant or lactating, among others [13].

### 2.2. HIV, HCV and HBV Serology

Data collection periods were divided into two sections, essentially because different serological test systems were in operation in the blood bank during each. Serological test systems used during the period 2012–2015 were: ARCHITECT anti-HIV/Agp24 (Combo), ARCHITECT Anti-HCV, ARCHITECT HBsAg Qualitative II and ARCHITECT Anti-HBc II, all of them from Abbott (Chicago, IL, USA). Samples with S/CO values > 1.0 are considered as reactive. Serological test systems used during the period 2017–2019 were: Laison XL Murex HIV Ab/Ag, Laison XL Murex HCV Ab (samples with S/CO values ≥ 1.0 are considered reactive), and Laison XL Murex HBsAg Quant (samples with UI/mL ≥ 0.050 are considered reactive), all of them from Diasorin (Saluggia, Italy). Data from July 2015 to December 2016 were not available and the present analysis corresponds to two non-continuous periods.

### 2.3. Individual Donor-Nucleic Acid Testing (ID-NAT) for HIV, HCV and HBV

Procleix Ultrio Assay HIV-1, HCV and HBV NAT assays were used on the TIGRIS Procleix System (Grifols Diagnostic Solutions Inc., Barcelona, Spain) to detect HIV-1 RNA, HCV RNA and HBV DNA in plasma or serum samples. Samples with S/CO values ≥ 1.0 were considered positive.

### 2.4. Statistics and Residual Risk Determination

Central tendency measures, Students’ *t*-test and ROC curves for serological test cut-off values to predict the probability of detecting RNA-HIV, RNA-HCV and DNA-HBV, were used. The estimation of residual risk (RR) for HIV, HBV, and HCV infections via blood transfusion was calculated according to the WHO guidelines [4]. Since in Mexico, in the years of study, the voluntary donation was estimated between 2.5 and 5% [12] all donors were considered as first-time donors (FTD), therefore incidence was used. HIV and HCV RR values were obtained multiplying incidence by the length of the viremic phase of the diagnostic window period (vDWP), defined as the period when at least 1 virus is present in a 20 mL plasma sample [4]. The lengths of the vDWP (in days) were selected in relation to the assay used in the years studied (Table 1). To calculate the RR for HBV, the HBV incidence adjustment factor was also obtained with the probability formulas (in %) of HBsAg detection, or the detection by ID-NAT-HBV [4].

### 2.5. Categorization of the Acute Viremia Phase (AVP) for HIV, HCV and HBV Infections

BD in HIV acute viremia phase (AVP-HIV) were defined as “Anti-HIV/p24Ag combo (−) and ID-NAT-HIV (+)” or “Anti-HIV (−), viral p24 Ag (−), and ID-NAT-HIV (+)” or “Anti-HIV (−), viral p24 Ag (+) and ID-NAT-HIV (−)” [4,14]. BD in HCV acute viremia phase (AVP-HCV) were defined as “Anti-HCV (−) and ID-NAT-HCV (+)” [4]. BD in HBV acute viremia phase were defined as “Anti-HBc (−), HBsAg (+) and ID-NAT-HBV (+)” or “HBsAg (−) and ID-NAT-HBV (+)” [15]. OBI subjects were defined as “Anti HBc (+), HBsAg (−), and ID-NAT-HBV (+)” (seropositive OBI) [16].

## 3. Results

### 3.1. Data Sources and Blood Donors

In 2011, ID-NAT tests were implemented at the IMSS CMN-MAC blood bank, which is in a large tertiary hospital in central Mexico. Four regional bleeding stations were gradually incorporated to perform the ID-NAT tests. Between January 2012 and June 2015, the serological and ID-NAT data of 49,792 BDs were collected. During a second period, from January 2017 to July 2019, the data of 56,333 BDs were collected, for a total of 106,125 individuals.

### 3.2. HIV, HCV and HBV Seroprevalence

During 2012–2015, HIV seroprevalence was 0.12% by the anti-HIV/p24Ag combo; while, during 2017–2019, when these markers were determined by separate assays, anti-HIV and p24Ag seroprevalences were 0.39% and 0.12%, respectively. In turn, the prevalence of anti-HCV in the first period was 0.34% increasing to 0.54% in the second. The prevalence of HBsAg increased ten-fold between the first and the second periods, from 0.10 to 1.03%. The anti-HBc prevalence was determined in the first period only, at 0.79% (Table 2).

### 3.3. ID-NAT Detection of HIV-RNA, HCV-RNA and HBV-DNA

HIV-RNA detection showed similar prevalence in the two periods, at 0.056% and 0.062%. HCV-RNA and HBV-DNA prevalence decreased in the second period from 0.058% to 0.026% for HCV, and from 0.060% to 0.044% for HBV (Table 2).

### 3.4. Blood Donors Detected in the Acute Viremia Phase (AVP) by Association of Serological Screening and ID-NAT

The main objective of NAT is to detect BD in the AVP of HIV, HCV or HBV infections, in the absence of seroconversion. ID-NAT testing detected 4 (1 HIV, 2 HCV, 1 HBV) and 6 (3 HIV, 3 HCV) BDs in the AVP, in the first and second periods, respectively, while 61 BDs in AVP were detected with p24Ag serology (Table 3).

### 3.5. Cut-Off Value of Reactive Serological Tests to Predict Viral RNA or DNA Detection with ID-NAT

Reactive serological tests do not necessarily imply that viral RNA or DNA will be detected by NAT. During the first period (2012–2015), positive ID-NAT-HIV correlated with anti-HIV/p24Ag combo reactivity with a predictive cut-off of S/CO = 13. In the second period (2017–2019), a cut-off value of S/CO = 23.9 was found for HIV RNA detection (Table 4). In turn, anti-HCV, in the first and second periods, had similar cut-off S/CO values of 4.35 and 4.45 for HCV RNA detection (Table 4). For HBV-DNA detection, the predictive S/CO cutoff values of serological markers were remarkably higher than their reactivity cut-offs, at S/CO = 10.4 for anti-HBc in the first period, and S/CO = 141 and 16.4 IU/mL for HBsAg, in the first and second periods, respectively (Table 4).

### 3.6. Determination of Residual Risk (RR) for HIV, HCV and HBV

The residual risk (RR) values were calculated considering the ID-NAT results in both periods, since the same technology was used in both (Table 1). The RR for HIV was 14 in 1 million blood donations, or 1 in 71,428, the RR for HVC was 6.8 in 1 million donations, or 1 in 147,058, and for HBV it was 156 in 1 million donations, or 1 in 6410 (Table 5).

### 3.7. Blood Donors with More Than One Marker during the Study Periods

In the period 2012–2015, a total of six individuals were found to be reactive or positive to more than one of the viruses. Among these cases, one donor showed reactivity for all three viruses through serology, but only tested positive for HIV and HBV through NAT. Additionally, three donors tested reactive for both HIV and HBV serology markers, but were only positive for HIV through NAT. The remaining three cases exhibited reactivity for HCV and HBV through serology, but only tested positive for HCV through NAT (Appendix A).

Similarly, in the period 2017–2019, a total of 20 individuals were identified as reactive or positive for more than one serological or molecular marker. Among them, seven donors showed reactivity for both HBV and HCV markers, but were only NAT positive for HCV. It is worth noting that all 20 donors tested positive for HBsAg, but only 2 of them tested positive for HBV through NAT (Appendix A).

## 4. Discussion

Despite highly sensitive technologies to ensure blood safety, the risk of infectious agent transmission through transfusion remains. To calculate this risk through time, it is important to periodically analyze the BD prevalence and markers of the most important viruses for transfusion medicine. In Mexico, there are limited studies that calculate these risks and there’s a lack of information by region, that would allow comparisons within and outside the country.

During the period covered in this study, another eight seroprevalence studies for HIV, HCV or HBV in Mexican BDs were published (Appendix A). Our seroprevalence, with the anti-HIV/p24Ag combo test, was 0.12% (Table 2), within the range reported by six of those studies: 0.13% (CI 95%: 0.092–0.16) (Appendix A) [10,17,18,19,20,21]. However, in the second period, HIV seroprevalence increased, likely because most anti-HIV/p24Ag combo assays produce a binary positive or negative result, without identification of the component causing reactivity [14]. In the current study (second period), anti-HIV produced more reactivity than p24Ag (Table 2).

Anti-HCV seroprevalence, in the first period, was below that reported by six of the studies conducted in Mexico, which, overall, averaged 0.50% (CI 95%: 0.44–0.56) (Appendix A). In the second period, anti-HCV seroprevalence increased and remained within the confidence interval of those studies. Despite this increase, the two anti-HCV seroprevalences were below 0.84%, which is the previously reported HCV prevalence for the state of Puebla [22]. These values are consistent with the better specificity and sensitivity of serological tests in detecting anti-HCV [23] and/or a better screening of BDs [13].

HBsAg seroprevalence, in the first period (0.10%, Table 2), was higher than previously reported for 2003–2009 (0.066%) [24], but within the confidence interval of seven blood banks in Mexico in the same period (Appendix A), which was calculated as 0.15% (95% CI 0.11–0.20) [10,17,18,19,20,21,25]. In the second period, we observed a ten-fold increase in this HBsAg seroprevalence, raising to 1.03%, perhaps due to an extra step in the assay, which now induced denaturation of the HBV core protein to expose the “a” determinant for interaction with the system’s murine anti-HBsAg monoclonal antibody (DiaSorin, Saluggia, Italia). However, the observed increase in seroprevalence could also be due to an actual increase in the number of cases. This is something that must evaluated in the coming years, both through the analysis of blood reserves and with monitoring to detect increases in the number of patients that require treatment. Furthermore, anti-HBc determination is not mandatory in Mexico. Rather, it is considered an additional test in BDs [13] and only a few blood banks report it [24,26]. In our study, anti-HBc was only determined in the first period with a seroprevalence of 0.79% (95% CI: 0.71–0.87) (Table 2). Both HBV serological markers were more prevalent than HIV and HCV serological markers.

Interestingly, among the three viruses studied, only HBV has a vaccine available, yet it was the most prevalent in the sample. The HBV vaccine has been included in Mexico’s national vaccination program since 1999. According to data up to 2017, vaccination coverage for HBV with three doses in newborns, has been reported between 84 and 89% [27,28]. However, the coverage percentages vary in older individuals. For example, it has been determined that 52% of health workers have been vaccinated at least once, but only 5.5% have complied with the complete scheme [29]. In the case of medical students, it was reported that 54% from a major university in Puebla had received at least one dose of the HBV vaccine, and more than 90% of those who received 2 or 3 doses had protective levels of anti-HBsAg [30]. Therefore, ideal levels of vaccine coverage have not yet been achieved in our population contributing to high prevalence, which could, in part, explain why the risk of iatrogenic transmission that we report here did not decrease further with NAT.

A considerable proportion (>80%) of BDs that were anti-HCV reactive were NAT-HCV negative (Table 2), as has been observed in Mexico for over a decade [18,21,22]. In marked contrast, 35% of BDs were only reactive to anti-HCV and NAT-HCV negative in France [31]. This could be due to Mexican BDs that naturally cleared the virus [32], or the serology assay could be cross-reacting, for example detecting antibodies to other flaviviruses [33] present in several Mexican regions [34]. These hypotheses remain to be explored. Nevertheless, the anti-HCV cut-off value (S/CO) to predict HCV-RNA detection (Table 4) has remained similar to that reported more than a decade ago using different NAT technologies [22,35].

NAT’s main use is to detect BD in the AVP of HIV, HCV and HBV infections that have not seroconverted. Over the periods studied, we found a rate of four HIV-AVP and five HCV-AVP in 105,768 donations (Table 3), similar to those reported in other Mexican studies in the same period (Table 6) [11,17,21]. Regarding OBI, we only detected one case in 52,574 donations using ID-NAT-HBV (Table 3). Recently, González-Santos et al. detected 1 in 66,137 in Mexico City [10] (Table 6), Dodd et al., in the USA, detected 1 in 67,974 [36] and Nishiya et al. reported 1 in 33,121 donations in Brazil [37].

Our RRs calculated for HIV, HCV and HBV after ID-NAT, are similar to a recent report from Lithuania, where ID-NAT tests decreased the RR compared to serological tests alone [38]; however, our results are different to the RRs reported for the US [36], where 73% of BDs were repeat donors. In Mexico, repeat altruistic BDs are only 2.5–5% [12], therefore first-time donors are above 95% and reactivity or positivity is higher in them compared to altruistic BDs [3,36,38].

Vazquez et al., in 2006, estimated that NAT implementation would reduce the RR to 1 HIV in 19,939 donations, 1 HCV in 9950 donations and 1 HBV in 8170 donations [39]. Our data show that the RR for HIV and HCV decreased more than projected but, for HBV, the risk remained slightly higher than expected (Table 5). Therefore, the overall conclusions are that NAT increased the safety of blood reserves significantly for HIV and HCV; however, it will be necessary to explore the conditions that prevented a larger decrease in the risk of HBV over the study period.

Due to their public health importance, Mexican public institutions have established programs for the treatment of HBV, HCV and HIV infections. The Mexican government has launched the hepatitis C elimination program, in alignment with the World Health Organization’s initiative, which encompasses infection screening and the availability of direct-acting antiviral treatments [40,41]. Furthermore, there are plans to develop a strategy for hepatitis B with the same objective in the near future [41]. These tasks are urgent and require medical systems with good population coverage and enhanced methods and algorithms for chronic viral infection detection and optimal treatment. By doing so, they can effectively contribute to the elimination of these infections [42,43]. Currently, nucleotide analogues are used as the first-line treatment for HBV, offering a high barrier to resistance [44]. In the case of HIV, patients receive treatment from public institutions. For instance, at the Mexican Institute of Social Security, the country’s largest public health institution, over 80,000 people living with HIV were under care by the end of 2021, with 97.4% of them receiving antiretroviral treatment. Among these patients, 91% achieved viral loads of less than 1000 copies, indicating treatment effectiveness [45]. These data highlight the existence of comprehensive programs with a significant impact on the Mexican population. However, it remains crucial to maintain and expand these programs to include more individuals and to enhance the efforts towards the HBV eradication plan in accordance with WHO guidelines.
viruses-15-01331-t006_Table 6Table 6Prevalence of HIV, HCV and HBV in BDs carried out in México since the introduction of NAT and during the period covered by the present study (2012–2019).
Reference

[11][8][9][17][18][10][46][21]TotalYear of publication200920102011201620172019202120212009–2021Study period2009–20102009–20102008–20092014–20152012–20152016–20192008–20182018–20192009–2019InstitutionINCCETSIMSSCETSISSSTEIMSSINPIMSS5Federal entityMexico CityJaliscoJaliscoJaliscoMexico CityNuevo LeónMexico CityJalisco3 of 32Number of BD19,062593847,84737,99936,793188,70564,98280,391481,717PD-NATYes, pool of 6Yes, pool of 6NoYes, pool of 6NoNoYes, pool of 6No4ID-NATNoNoYesNoYesYesYes (2018)Yes5Prevalence by NAT, N (P%)Mean prevalence (CI 95%) ^a^HIV6 (0.031)0 (0.0)16 (0.033)10 (0.026)18 (0.049)NR18 (0.027)32 (0.039)0.034 (0.025–0.043)HCV8 (0.041)5 (0.08)56 (0.11)44 (0.11)22 (0.059)NR22 (0.033)45 (0.055)0.069 (0.040–0.098)HBV18 (0.094)7 (0.11)26 (0.054)11 (0.028)12 (0.033)NR7 (0.010)20 (0.025)0.050 (0.015–0.085)Blood donors in acute viremia phase, N (R) ^b^HIV1 (1:18,944)001 (1:37,933)NR2 (1:94,245)01 (1:80,239)5 (1:96,343)HCV2 (1:9971)1 (1:5938)04 (1:9488)NR1 (1:187,788)04 (1:20,001)12 (1:40,143)HBV1 (1:18,732)000NR2 (1:94,224)02 (1:40,105)5 (1:96,343)








22 (1:21,896)OBI0000NR3 (1:66,137)00 (0.0)3 (1:160,272)INC = National Institute of Cancerology, CETS = State Blood Transfusion Centers (Jalisco), IMSS = Mexican Institute of Social Security (States Jalisco and Nuevo León), ISSSTE = Institute for Social Security and Services for State Workers (Mexico City), National Institute of Pediatrics; ^a^ = mean prevalence was calculated as simple mean of prevalence values by year; ^b^ = Ratio of BD in acute phase respect to those of negative serology.


Our study has certain limitations that should be taken into consideration. Firstly, it is a monocentric study, although it includes donors from four blood collection stations in the state of Puebla (Tehuacan, Teziutlan, Atlixco and Angelopolis). Approximately 75% of the samples come from the state’s central region (Angelopolis), where the state capital Puebla City is located. Secondly, we were unable to analyze donation results throughout the entire study period (data for the year 2016 are absent from the study). These limitations highlight the need for future studies that encompass larger geographical areas, which may present different risk factors and variations in access to healthcare systems. Additionally, efforts should be made to address any challenges regarding data accessibility, as these data play a crucial role in research.

We should not overlook the fact that a proportion of the donors tested positive or were reactive for one or more of the three viruses (Appendix A). It is important to emphasize that beyond the individual infection status, these findings underscore the significance of conducting both serological and molecular tests to minimize the risk of iatrogenic transmission in blood banks. Since multiple tests provide confirmatory evidence and complement each other, ID-NAT implementation is crucial for ensuring the safety of blood reserves.

## Figures and Tables

**Table 1 viruses-15-01331-t001:** Length of the viremic phase of the diagnostic window period (vDWP) for HIV, HCV and HBV of different tests used in a blood bank in Puebla City, Mexico between 2012 to 2015 and 2017 to 2019.

Virus	Years	Screening Assay	Manufacturer	vDWP (Days)	vDWP (Year)
HIV	2012–2015	ChLIA (Anti-HIV/p24Ag)	^a^ ARCHITECT HIV Ag/Ab Combo	16	0.0438
	2017–2019	ChLIA (Anti-HIV)	^b^ Laison XL Murex HIV Ab/Ag	21	0.0575
	2017–2019	ChLIA (p24Ag)	^b^ Laison XL Murex HIV Ab/Ag	14	0.0383
	2012–2019; 2017–2019	ID-NAT (HIV-RNA)	^c^ Procleix Ultrio Assay	8	0.0219
HCV	2012–2015	ChLIA (Anti-HCV)	^a^ ARCHITECT Anti-HCV assay	60	0.1642
	2017–2019	ChLIA (Anti-HCV	^b^ Laison XL Murex HCV Ab	60	0.1642
	2012–2015; 2017–2019	ID-NAT (HCV-RNA)	^c^ Procleix Ultrio Assay	5	0.0136
HBV	2012–2015	ChLIA (HBsAg)	^a^ ARCHITECT HBsAg	42	0.1150
	2017–2019	ChLIA (HBsAg)	^b^ Laison XL Murer HBsAg Quant	42	0.1150
	2012–2015; 2017–2019	ID-NAT (HBV-DNA)	^c^ Procleix Ultrio Assay	27	0.0739

Calculations were carried out following the WHO guidelines (OMS). ChLIA = Chemiluminescence immunoassay; ID-NAT = Individual nucleic acid amplification technique; Combo = Simultaneous detection of Anti-HIV and p24Ag; vDWP = viraemic phase of the diagnostic window period; ^a^ Abbott Laboratories North Chicago, IL, USA; ^b^ Diasorin, Saluggia, Italy; ^c^ Norvartis, Basel, Switzerland.

**Table 2 viruses-15-01331-t002:** Prevalence of serological and ID-NAT tests for HIV, HCV and HBV in a blood bank in Puebla City, Mexico, between 2012–2015 and 2017–2019.

	2012–2015			2017–2019		
Serological Test or ID-NAT	BD (+) N	BD (−) N	Prevalence (CI 95%)	BD (+) N	BD (−) N	Prevalence (CI 95%)
HIV						
Anti-HIV/p24Ag	62	49,730	0.124 (0.10–0.16)	NP	NP	-
Anti-HIV	NP	NP	-	225	56,108	0.39 (0.34–0.45)
p24Ag	NP	NP	-	72	56,261	0.12 (0.10–0.15)
ID-NAT-HIV	30	49,762	0.056 (0.04–0.08)	35	56,298	0.062 (0.044–0.080)
HCV						
Anti-VHC	168	49,624	0.34 (0.28–0.38)	309	55,989	0.54 (0.48–0.60)
ID-NAT-HCV	29	49,763	0.058 (0.037–0.079)	15	56,318	0.026 (0.022–0.030)
HBV						
HBsAg	50	49,743	0.10 (0.071–0.12)	584	55,749	1.03 (0.95–1.11)
Anti-HBc	396	49,396	0.79 (0.71–0.87)	NP	NP	-
ID-NAT-HBV	32	49,760	0.064 (0.042–0.086)	25	56,308	0.044 (0.027–0.061)
Total, serology	675	49,114	1.36 (1.26–1.46)	1120	55,213	1.98 (1.87–2.10)
Total, ID-NAT	91	49,701	0.18 (0.16–0.22)	75	56,258	0.13 (0.10–0.16)

ID-NAT = Individual determination nucleic acid test; BD = blood donors; NP = Not performed in the period; (−) Not calculated; prevalence values (%) of serological and NAT tests were based in total blood donors in each period (49,792 and 56,333, respectively).

**Table 3 viruses-15-01331-t003:** Blood donors in the acute viremic (AVP) phase of HIV, HCV and HBV estimated by the association of screening serology and ID-NAT.

2012–2015					2017–2019				
Algorithm	BD N	Phase of Infection	Serology (S/C0)	Prevalence (%)	Algorithm	BD N	Phase of Infection	Serology (S/C0)	Prevalence (%)
HIV									
Anti-HIV/p24Ag Combo (−), ID-NAT HIV (+)	1	AVP-HIV	0.12	0.002	Anti-HIV (−), p24Ag (+), ID-NAT HIV (−)	61	AVP-HIV	3.20	0.110
					Anti-HIV (−), p24Ag (−), ID-NAT HIV (+)	3	AVP-HIV	0.23, 0.38	0.005
HCV									
Anti-HCV (−), ID-NAT HCV (+)	2	AVP-HCV	0.420	0.004	Anti-HCV (−), ID-NAT-HCV (+)	3	AVP-HCV	0.41	0.005
HBV									
HBsAg (+); Anti-HBc (−); ID-NAT HBV (+)	1	AVP-HBV	30.2; 0.02	0.001	HBsAg (−), ID-NAT HBV (+)	0	AVP-HBV		0.000
HBsAg (−), Anti-HBc (−); ID-NAT HBV (+)	0	AVP-HBV		0.000					
HBsAg (−), anti-HBc (+), ID-NAT HBV (+)	2	OBI	0.17; 12.10	0.002					

BD = blood donors; S/CO = Analyte signal cutoff; (+) = Screening serology reactive anti-HIV/p24Ag = S/CO ≥ 1.00; (−) = Nonreactive screening serology anti-HIV/p24Ag combo = S/CO < 1.00; ID-NAT-HIV = Procleix HIV-1 discriminatory assays; (+) = Positive ID-NAT-HIV test S/CO ≥ 1.00, (−) = Negative ID-NAT-HIV test S/CO < 1.00; (+) = screening serology reactive anti-HIV or p24Ag = S/CO ≥ 1.00; (−) = nonreactive screening serology anti-HIV or p24Ag = S/CO < 1.00; AVP = acute viremic phase for HIV, HCV and HBV; AVP prevalence was calculated based in BDs not reactive to serological test in each period for HIV, HCV and HBV; OBI = occult HBV infection.

**Table 4 viruses-15-01331-t004:** Cut-off values of serological tests to predict the probability of detecting HIV-RNA, HCV-RNA and HBV-DNA by correlating levels of reactive serological test with levels of positive ID-NAT (ROC curve) in blood donors from Puebla, México.

NAT		Serological Test	Reactive Serological Tests (S/CO)	S (%)	E (%)	Serological Test Associated with Detecting RNA or DNA (S/CO)	AUC
HIV	1st period	Combo anti-HIV/p24Ag	≥1.0	90 (73.5–97.9)	94.4 (81.4–99.3)	13.0	0.934
	2nd period	Anti-HIV	≥1.0	85.7 (69.7–98.4)	98.4 (95.6–99.7)	23.9	0.864
HCV	1st period	Anti-HCV	≥1.0	82.1 (63.1–93.9)	94.6 (90.4–97.4)	4.35	0.832
	2nd period	Anti-HCV	≥1.0	80.2 (61.2–99.2)	97.1 (94.2–99.6)	4.45	0.791
HBV	1st period	HBsAg	≥1.0	92.6 (75.7–99)	100 (82.2–100)	141	0.925
		Anti-HBc	≥1.0	96.7 (83.3–99.9)	82.7 (78.4–86.4)	10.1	
	2nd period	HBsAg	≥0.05 UI/mL	100 (86.3–100)	99.8 (99–100)	16.4 UI/mL	1.00

First period (2012–2015); second period (2017–2019); S = sensitivity; E = specificity; S/CO = analyte signal cut-off; AUC = area under the ROC curve.

**Table 5 viruses-15-01331-t005:** Residual risk of HIV, HCV or HBV infections in blood donations for serological and ID-NAT tests.

	Donations (N)	RR per Million Donations (Combo)	RR per Million (Antibody)	RR per Million (Antigen)	RR per Million Donations (ID-NAT)
HIV					
2012	8191	59	NP	NP	20
2013	9246	85	NP	NP	25
2014	14,156	55	NP	NP	11
2015	18,199	43	NP	NP	3.4
2017	22,110	NP	293	58	11
2018	19,566	NP	199	48	15
2019	14,657	NP	172	31	13
Average		60.2	221	45.6	14
HCV					
2012	8191	NA	740	NA	11.5
2013	9246	NA	833	NA	13.3
2014	14,156	NA	695	NA	10.6
2015	18,199	NA	462	NA	1.5
2017	22,110	NA	1127	NA	2.4
2018	19,566	NA	1064	NA	4.2
2019	14,657	NA	727	NA	4.6
Average			806		6.8
HBV					
2012	8191	NA	NA	121	31
2013	9246	NA	NA	1615	477
2014	14,156	NA	NA	1547	202
2015	18,199	NA	NA	547	43
2017	22,110	NA	NA	11,573	141
2018	19,566	NA	NA	9617	80
2019	14,657	NA	NA	9374	124
Average				4913	156

Combo = simultaneous detection of antibody and antigen; NP = not performed in the period; NA = not applicable. All donations were considered first time donations in that blood bank because they were replacement donations, not altruistic donations.

## Data Availability

Not applicable.

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
