# Peer review of "Donated Blood Screening for HIV, HCV and HBV by ID-NAT and the Residual Risk of Iatrogenic Transmission in a Tertiary Care Hospital Blood Bank in Puebla, Mexico"

_viruses, 2023, doi:10.3390/v15061331_

Round 1

Reviewer 1 Report

Dear authors, I read your paper, and I found it very easy to read and interesting. Data are well presented, results and clear, and the discussion is appropriate, too. I suggest to accept the paper after few revision.

Main issue: please include in the discussion a "limits of the study" paragraph, including that it is a monocentre study and any other study limitations you can detect;

Minor issue: Just a question about HBV: despite you give an appropriate laboratory/methodological reason for increase of HBV, you said the a real increase of cases cannot be excluded. I suggest to integrate this point with some information about HBV vaccination policies and coverage in MExico. In most EU countries, HBV vaccination is mandatory since the mid 80ies of last century, so HBV is progressively disappearing. What about HBV vaccination in Mexico) 

Reviewer 2 Report

This is a large cross-sectional study of blood-borne pathogens in blood donors across two distinct study periods in Puebla, Mexico.   Overall, the manuscript is well written, and the methodologies are appropriate and well described.  Minor revisions would strengthen the manuscript further including:

·      Inclusion criteria are mentioned in section 2.1 but should be stated explicitly.

·      How many individuals had more than infection diagnosed simultaneously?

·      Are there data to suggest subtype- or genotype-specific NAT detection rates for HIV, HBV or HCV?

·      The discussion should include currently HBV vaccination standards, as well as HIV, HBV, and HCV treatment options available in both the public and private sections (if different).
